# Aligning to the UN Sustainable Development Goals: Assessing Contributions of UBC Botanical Garden

Adriana Lopez-Villalobos *, Dionne Bunsha, Delanie Austin, Laura Caddy, Jennifer Douglas, Andy Hill, Kevin Kubeck, Patrick Lewis, Ben Stormes, Ryo Sugiyama and Tara Moreau *

UBC Botanical Garden, Faculty of Science, University of British Columbia, Vancouver, BC V6T 1Z4, Canada; dionne.bunsha@ubc.ca (D.B.); delanie.austin@ubc.ca (D.A.); laura.caddy@ubc.ca (L.C.); jennifer.douglas@ubc.ca (J.D.); andy.hill@ubc.ca (A.H.); kevin.kubeck@ubc.ca (K.K.); patrick.lewis@ubc.ca (P.L.); ben.stormes@ubc.ca (B.S.); ryos@mail.ubc.ca (R.S.)
* Correspondence: adriana.lopez@ubc.ca (A.L.-V.); tara.moreau@ubc.ca (T.M.)

**Abstract:** The United Nations 2030 Agenda for Sustainable Development outlines 17 goals for the wellbeing of people and the planet. The purpose of this study was to understand how University of British Columbia Botanical Garden (UBCBG) contributes to the United Nations Sustainable Development Goals (UN-SDGs) and to identify opportunities for future action. To address this, we worked across departments to assess our programs and activities against the UN-SDG 17 goals and 169 targets. The UN-SDG indicators were only used to identify potential metrics that could be consider for future tracking. The main activities of UBCBG include ex situ plant conservation, sustainability education and community engagement. Our results found that UBCBG contributes to 12 of the 17 goals and 24 of the 169 targets. The two UN-SDGs with more targets aligned to UBCBG's activities were Goal 15—Life on Land and Goal 12—Responsible Consumption and Production. Through its partnerships with other botanical gardens, research institutions and the regional government, the Garden amplifies its work at a global, national and regional level. We are re-imagining the role of botanical gardens in an age of equity, decolonization, the biodiversity crisis and the climate emergency. Since the UN-SDGs address both nature and people, they are an appropriate framework to guide our work.

**Keywords:** botanical gardens; environmental education; plant conservation; sustainability; United Nations Sustainable Development Goals; university botanical garden

## 1. Introduction

Biodiversity, and especially plant diversity, is being lost at an unprecedented rate. It is estimated that two in five plant species face extinction [1] and three-quarters of the land-based environment plus about 66% of the marine environment have been significantly altered by human actions [2]. Future climate change projections offer a dire view of what future generations will face [3]. The urgency of both the biodiversity crisis and climate emergency requires our immediate attention. Botanical gardens (and arboreta) are at the forefront of plant conservation and are well equipped to tacklethese crises with their technical expertise, facilities and plant science research. In addition, the global network of over 2500 botanical gardens stewardsnearly 6.1 million living plant accessions, representing about one-quarter of the estimated number of vascular plant species) [4–8]. Botanical gardens also play a central role in meeting human needs, such as providing green spaces, aiding to the well-being of people and developing public educational programing to raise awareness of environmental issues and climate change [9]. Many of these activities are part of a garden's mission and can be important contributions to sustainable development [10,11].

The United Nations' Sustainable Development Goals (UN-SDGs) are a call to action to protect the planet and ensure a sustainable future for all. Signed in 2015, they outline a global policy and plan for the 2030 Agenda for Sustainable Development, which aims to

advance sustainability across social, environmental, economic and governmental agendas. Developed to replace the Millennium Development Goals, the UN-SDGs describe 17 goals, 169 targets and 232 indicators over a 15-year time frame. Critiques of the UN-SDGs highlight a lack of focus on Indigenous Rights [12,13] as well as limited attention to topics such as degrowth [14,15]. Yet, despite their limitations the UN-SDGs are a globally important framework currently guiding and mobilizing sustainability approaches across political, business and community scales.

The University of British Columbia Botanical Garden (UBCBG) is located on the ancestral and unceded lands of the Musqueam People, in the southwestern corner of British Columbia on the Pacific coast of Canada. UBC Botanical Garden comprises three sites on the University of British Columbia's Point Grey (Vancouver) campus: Botanical Garden, Nitobe Memorial Garden and Botanical Garden Nursery. The Botanical Garden consists of several named garden areas and an area of undeveloped second-growth forest.

Looking back on its 100+ year history reveals a complex and variable relationship with sustainability. UBCBG's first director, John Davidson (1916–1948), was an advocate for school gardens, local watershed conservation and plant conservation. Throughout its early history, garden leadership and university directions focused on plant collecting and horticulture with varying emphasis on sustainability. In April 2010, *Redefining the Role of Botanic Gardens—Towards a New Social Purpose* [16] called on gardens to work within their communities to address concerns such as climate change. This inspired the UBCBG to address issues of sustainability and human impacts on nature. In 2014, the Garden created a sustainability and community programs (formerly education) department and began the process of reaching out to different audiences, including businesses.

In 2015, The Sustainable Communities Field School (Field School) was established with donor funding at UBCBG. Through team building tours, the Field School engaged businesses in learning about the City of Vancouver's Greenest City Action Plan and its 10 goal areas. In 2017, alignment to the UN-SDGs extended the curriculum beyond local government policies to the global setting. Collaborations with behavioral science faculty, graduate students and a community non-profit (Society Promoting Environmental Conservation) helped inform experience design and delivery. Behavioral science research evaluated the impact of the Field School on environmental knowledge, attitudes and willingness to engage in pro-environmental behaviors [17]. In addition to research, a key outcome of the Field School was an online toolkit designed to support botanical garden educators in integrating the UN-SDGs into garden tours and experiences [18]. For UBCBG, UN-SDG programming was inspired and informed by the growing number of university initiatives and botanical garden activities [19,20].

In early 2021, UBCBG embarked on a multi-year initiative to advance climate adaptation planning, develop a regional biodiversity atlas and further deliver climate education. However, many questions remain. For example, how do we effectively achieve our mission to conserve and document plants and support our communities to respond to climate change and live sustainably? What is the best way to track and evaluate the progress towards being a more sustainable Garden? The UN-SDGs represent an interdisciplinary system to advance wellbeing, ecosystem services, cultural diversity and life on earth. As a global framework, they address high-level goals (17), targets (169) and indicators (232). In this paper, we explore which UN-SDG goals, targets and indicators the UBCBG is currently contributing to and where opportunities for future action might be.

## 2. Materials and Methods

In 2015, UBCBG's Field School began monitoring garden activities against the UN-SDGs with a particular focus on public education and community engagement. Sustainability activities and policies set by UBC (e.g., UBC Climate Action Plan 2030 [21]) provide institutional support for promoting UN-SDGs with uptake and interest from students, faculties and staff. Increased awareness and understanding of how UBC and UBCBG contribute to the UN-SDGs provide an important baseline of action as well as highlighting

promising future directions. 'Leaving no one behind' is a key element to the inclusivity of the UN-SDGs and increasing the capacity of UBCBG staff to understand the UN-SDGs is important to achieving the targets.

The *Sustainability* Special Issue—Botanic Gardens and Their Contribution to the Sustainable Development Goals—provided UBCBG with an opportunity to collaboratively assess its progress towards the UN-SDGs. By working across Garden departments (Administration, Collections and Horticulture, Sustainability and Community Programs, and Operations), we developed an evaluation framework to assess UBCBG contributions by reviewing our programs and activities and mapping them to the UN-SDGs. The process started with a review of the 17 goals, 169 targets and 232 indicators. In line with previous research, we found that most indicators were difficult to apply to organizations [22]. Therefore, the UN-SDG indicators were not included in our evaluation framework, and garden activities were assessed only for the goals and targets. However, the UN-SDG indicators were used to guide the development of potential metrics that UBCBG could consider for future tracking and evaluation (Supplementary Materials File S1). This was done to contribute towards the establishment of garden-based indicators that could be used for benchmarking.

Inspired by plant conservation and biodiversity benchmarking tools (e.g., Public Gardens Sustainability Index, Conservation and Biodiversity Benchmarking) created by the American Public Gardens Association (APGA), Botanic Garden Conservation International (BGCI) and the United States Botanic Garden, we assessed UBCBG's progress towards the UN-SDGS by answering the following questions:

(1)     To which goals are UBCBG contributing?
(2)     To which targets are UBCBG contributing?
(3)     What garden-scale metrics could UBCBG potentially use to evaluate and track progress towards targets and goals?
(4)     Which are the key goals, targets or indicators in which the UBCBG requires further attention and action?

To explore and answer these questions, UBCBG programs and activities were identified by staff from across the organization. When specific programs or activities contributed to more than one goal and target, the goal deemed most relevant for a program was selected to avoid duplication. For each goal, relevant targets were identified for specific UBCBG activities, and the number of targets being addressed was tallied. Through this process, we also identified the goals not being addressed by UBCBG, which provides information about potential programming gaps as well as highlighting opportunities for future action.

This study was a self-evaluation of the garden against the SDGs conducted by staff at the UBCBG and may inherently contain some bias. However, we tried to be as impartial and objective as possible and aimed to follow a SWOT (Strengths, Weaknesses, Opportunities and Threats) assessment to provide as holistic a review as possible.

## 3. Results

### 3.1. UBCBGs Contributions to UN-SDG Goals and Targets

Through its programs and activities, UBCBG contributed to 12 of the 17 UN-SDGs goals and 24 of 169 targets (Figure 1; Table 1). A total of 32 contributions to the UN-SDGs from UBCBG were identified from the programs and activities reviewed. The two goals that UBCBG contributes to the most were Goal 15—Life on Land and Goal 12—Responsible Consumption and Production, with five and four targets aligned, respectively. For Goals 2, 4, 6, 13 and 17, we identified contributions to two targets, and for Goals 3, 7, 8, 10 and 11 only for one target. We did not identify UBCBG contributions for any of the targets of Goals: 1—No Poverty, 5—Gender Equality, 9—Industry, Innovation and Infrastructure, 14—Life below water and 16—Peace, Justice and Strong Institutions (these were excluded from Table 1, but they can be found in Supplementary Materials File S1).

**Table 1.** UBCBG's contributions to UN-SDGs goals and targets with potential garden-based metrics. The proposed metrics are expressed in relative values as these are more effective and relevant for internal evaluation purposes; however, in some cases absolute values are included.

| UN-SDG | UN-SDG Targets | UBCBG Contributions | Potential UBCBG Metrics |
|---|---|---|---|
| **2. Zero Hunger**<br> | 2.4 | Growing and distributing fruits and vegetables to students in need, food banks and food security programs. | Proportion of area in food production and weight of food harvested and distributed annually (relative to the total available area for food cultivation and the highest weight harvested across years) |
| | | Delivering educational programs on food security, sustainable agriculture and food waste reduction. | Percentage of food related programs or tours delivered annually (relative to all educational programs). |
| | 2.5 | Conserving food plant diversity and crop wild relatives (CWR) through ex situ living collections. | Percentage of crop wild relative (CWL) accessions alive in the collection by genus. |
| | | Building capacity for conserving food plant diversity and crop wild relative conservation in United States and Canada by collaborating in the development of a conservation road map. | Actions taken towards implementing the road map for crop wild relative conservation in the US and Canada.<br>Number of conservation plans implemented relative to the total taxa identified from the road map. |
| **3. Good Health and Well Being**<br> | 3.4 | Partnering with UBC Wellbeing and other groups (e.g., a Prescription for Nature—PaRx) to increase access to nature and education. | Percentage of nature prescription access passes provided annually since program implementation. |
| | | Conducting mental health and wellbeing focused group tours. | Percentage of participants on wellbeing tours (relative to all participants from all educational programs or tours yearly). |
| **4. Quality Education**<br> | 4.4 | Training youth and adults with horticulture skills graduating from the Horticulture Training Program (HTP). | Percentage of students graduating from HTP each year since 2012 (relative to students enrolled in the program each year and across years). |
| | 4.7 | Delivering educational experiences that raise awareness of the UN-SDGs and link plant conservation with cultural diversity. | Percentage of participants on educational experiences (relative to all participants from all educational programs or tours yearly). |
| | | Installing interpretative signage to include Indigenous knowledge and languages. | Percentage of display signs that include Indigenous Languages in the Garden. |
| | | Promoting a culture of peace, global citizenship, and appreciation of cultural diversity through the Nitobe Memorial Garden, a traditional Japanese stroll garden. | Percentage of visitors to the Nitobe Memorial Garden annually (relative to all visitors to UBCBG yearly). |

**Table 1.** *Cont.*

| UN-SDG | UN-SDG Targets | UBCBG Contributions | Potential UBCBG Metrics |
|---|---|---|---|
| **6. Clean Water and Sanitation** | 6.4 | Conserving fresh water and improving water-use efficiency through irrigation upgrades using low-pressure irrigation heads, automated wireless valves and controllers. | Proportion of the garden area with updated irrigation infrastructure relative to the total cultivated area. |
| | | Planning garden space so irrigation is minimal after collection establishment. | Proportion of garden area with minimal irrigation needs relative to total area cultivated. |
| | | Providing education on the importance of water conservation through education and interpretive signs. | Percentage of participants engaged in water conservation tours and programs (relative to total participants in other programs). |
| | 6.b | Producing the Grow Green Guide [23] in collaboration with regional government to help local communities to install water efficient gardens. | Grow Green Guide website [23] visitor statistics. |
| **7. Affordable and clean energy** | 7.3 | Transitioning towards clean and sustainable sources of energy used in Garden operations through e-machinery and retrofitting. | Percentage reduction in energy consumption. Percentage of machinery that is using electric energy |
| **8. Decent work and economic growth** | 8.9 | Promoting sustainable tourism that creates jobs and promotes local culture and products by hosting local and global tourists in the Garden and on the Greenheart TreeWalk. | Percentage of visitors that are tourists (relative to total visitor population per year). |
| **10. Reduced inequalities** | 10.2 | Advancing workplace inclusion, diversity, equity, and accessibility through the implementation of Inclusion Diversity Equity and Accessibility (IDEA) at the Garden. | Number of actions implemented to advance IDEA annually (relative to actions planned for the year or compared to the previous year). |
| | | Increasing access to the Garden for low-income children, youth, and families. | Percentage of families accessing the Garden for free yearly (relative to total families entering the garden). |

**Table 1.** *Cont.*

| UN-SDG | UN-SDG Targets | UBCBG Contributions | Potential UBCBG Metrics |
|---|---|---|---|
| **11. Sustainable cities and communities** | 11.3 | Participating in local advisory groups that promote biodiversity mainstreaming within urban design and development. | Percentage of local committees and advisory groups that Garden staff participate in (relative to all potential groups that enhance inclusive and sustainable urbanization in Metro Vancouver). |
| **12. Responsible consumption and production** | 12.2 | Promoting the sustainable use of natural resources by reducing the use of plastics and peat at the Nursery when propagating plants. | Percentage of pots used made of more robust material that can be reused and recycled (relative to plastic pots currently in use and/or used in the past). |
| | 12.5 | Zero waste events are organized by the Garden to raise awareness about reducing and reusing materials. | Percentage of zero waste events organized (relative to the total number of educational events organized). |
| | 12.6 | The Sustainable Communities Field School engages businesses in team building experiences in the Garden to learn about the UN-SDGs as well as local sustainability action. | Percentage of sustainability tours for businesses (relative to all educational programs or tours). |
| | 12.8 | Education and public engagement for biodiversity conservation, sustainable development, and climate action through tours for school students and teachers, teambuilding experiences and public events. | Percentage of people who participated in educational tours for sustainable development (relative to the total participants from all educational programs or tours). |
| **13. Climate action** | 13.2 | Launching a 5-year climate adaptation planning process to increase capacity of UBCBG and other botanical gardens to adapt to climate change and reduce greenhouse gas emissions. | Number of actions taken to advance climate adaptation (relative to those stablished in the plan). |
| | 13.3 | Mobilizing youth towards the UN-SDGs through the UNLEASH Innovation Lab. In 2019, a UBCBG staff and student participated in UNLEASH in Shenzhen China with 1000 youth + 200 facilitators collaborating on solutions to reach the UN-SDGs. | Youth engaged in building capacity on UN-SDGs and climate change action. |

**Table 1.** *Cont.*

| UN-SDG | UN-SDG Targets | UBCBG Contributions | Potential UBCBG Metrics |
|---|---|---|---|
| **15. Life on land** | 15.1 | Through partnerships with BGCI Consortia and APGA Plant Collections Network, as well as Index Seminum (botanical garden to botanical garden seed exchange), the Garden is conserving threatened groups of plant species. | Percentage of species and existing accessions per group backed-up in UBCBG collection. |
| | 15.4 | Ex situ collection of montane plants from different ecosystems around the world. | Percentage of montane taxa and accessions of known wild origin in the collection from different ecosystems around the world (relative to the total number of montane taxa and accession in the garden). |
| | 15.5 | Ex situ collection of International Union for Conservation of Nature (IUCN) red-listed species. | Percentage of IUCN red-listed species in the collection. |
| | 15.8 | Providing expertise and knowledge on regional invasive species. | Number of invasive species factsheets developed (relative to total planned). |
| | 15.a | Maintaining a university botanical garden that works towards plant conservation, education, research and engagement. | Annual operating budget and financial growth over time. |
| **17. Partnerships for the goals** | 17.16 | Collaborating with botanical gardens and networks across local, regional and international scales to advance plant conservation. | Number of botanical gardens and number of garden networks that the UBCBG collaborates with (relative to total locally and world-wide). |
| | 17.17 | Partnering with local and regional governments in Metro Vancouver to build capacity for sustainable horticulture. | Number of partnerships with government (relative to all potential partnerships). |

The results of the assessment indicate examples of UBCBG's contributions to the UN-SDGs as well as gaps in our progress. Additional information about UBCBG's contributions to the goals and targets (in brackets) is described below.

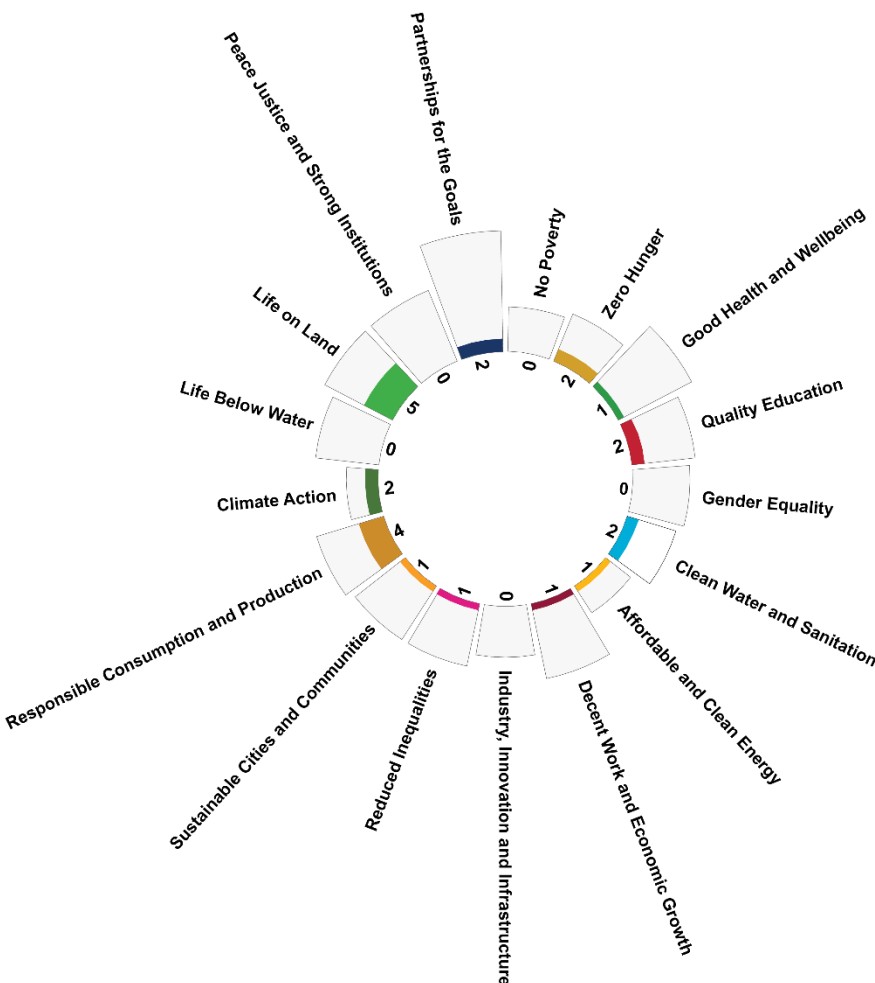

**Figure 1.** UBCBG's contribution to the 17 UN-SDGs and 169 targets. The height of each bar is proportional to the number of UN-SDG targets for that goal. The colored portion of the bar represents the number of targets that were mapped to UBCBG programs and activities contributing towards that goal and the numbers inside of the circle are those targets. The height of the bar represents the total number of targets set by the UN-SDGs for each of the 17 goals in clockwise direction, starting with Zero Poverty, and are as follows: 7, 8, 13, 10, 9, 8, 5, 12, 8, 10, 10, 11, 5, 10, 12, 12, 19 (total UN-SDG targets = 169).

### 3.2. Goal 15—Life on Land (Five Targets by UBCBG)

Goal 15 focuses on managing forests sustainably, restoring degraded lands and successfully combating desertification, reducing degraded natural habitats and ending biodiversity loss. For botanical gardens, Goal 15 is considered to be the most relevant to our work and is our key contribution to conservation [24]. By maintaining an ex situ collection of more than 5000 plant taxa and cultivars from almost 200 plant families, representing a wide range of ecosystems, UBCBG contributes to backing up plant diversity that might otherwise be lost in nature and in human-dominated ecosystems (15.1, 15.a). The garden holds several important plant collections that show different degrees of conservation concern or rarity in the wild (15.5). Key conservation collections include the genera Acer, Magnolia and Rhododendrons, Sorbus, Viburnum, Hydrangea, Cornus and Stirax (maples, magnolias, rhododendrons, Asian whitebeams and mountain ashes, viburnums, hydrangeas, dogwoods, as well as wide representations of the snowbell and lardizabala families). However, what makes a collection valuable from a conservation point of view are metacollections. These groups represent the collaborative efforts with other botanical institutions to manage combined plant collections, generate knowledge and develop strategies for ex situ plant

conservation. Metacollections increase coverage within plant groups in terms of number of species and genetic diversity represented in natural populations. When used for in situ conservation, metacollections have the capacity to boost species numbers and genetic diversity in natural populations. For individual institutions, they dilute the risk of loss and reduce maintenance costs [25].

In 2019 the Garden was chosen as the lead institution in the Global Conservation Consortium for Acer (GCCA), one of six global conservation consortia managed by Botanical Gardens Conservation International (BGCI). Beyond its own collections, UBCBG is also part of the Acer Multisite collection administered by the Plant Collections Network (PCN) of the American Public Garden Association (APGA). To date, there are thirteen North American botanical institutions that contribute to the PCN Acer multisite collection. In addition, the garden is part of the PCN's Magnolia Multisite collection and an active participant in the Global Conservation Consortia for Magnolias, Rhododendrons and Oaks. Through the Index Seminum program, the Garden participates in an exchange of seeds between botanical gardens and arboreta.

Particularly vulnerable are alpine ecosystems. These are habitats that suffer more strongly from the impacts of climate change [26,27]. Target 15.4 specifically refers to the conservation of mountain ecosystems, including their biodiversity, in order to enhance their capacity to provide benefits that are essential for sustainable development. The E.H. Lohbrunner Alpine Garden (Alpine Garden) at UBCBG holds a diverse collection of montane and alpine species from Asia, Africa, Europe, North and South America and Australasia. Although the number of true alpines—diminutive plants adapted to extreme cold, wind and a short growing season—is relatively low, the ability to showcase the nature and diversity of alpine ecosystems, made precarious by climate change, has considerable value.

### 3.3. Goal 12—Responsible Consumption and Production (Four Targets by UBCBG)

UBCBG's actions towards 'Responsible Consumption and Production' includes continual improvements to sustainable horticultural practices (12.2), educational tours that highlight sustainable consumption (12.6), team building activities for businesses (12.8) and zero waste events (12.5). Efforts to reduce single-use plastics in the nursery includes using pots made of more robust material that can be reused and ultimately recycled. This shift towards different containers reduced annual use of single-use plastics from ~5000 to about 300. Other improvements for sustainable horticulture include trialing peat moss alternatives as a means of reducing the use as a non-renewable resource. In the past 7 years, there has been a 50-fold decrease in the use of peat at the nursery. Today, coco fibre (coir) is effectively used as an alternative to peat moss.

The Field School contributes to multiple UN-SDGs and targets (Figure 2). Target 12.6, which focuses on encouraging companies to adopt sustainable practices, was a key focus for the program. The curriculum includes actions for sustainable food systems, water conservation, biodiversity and waste reduction. Groups were introduced to the UN-SDGs through team building activities conducted in the garden. Zero waste education and implementation were key foci of the Field School, with annual contributions to zero waste education and research at the UBCBG's Apple Festival (Figure 3).

### 3.4. Goal 2—Zero Hunger (Two Targets by UBCBG)

UBCBG has been operating the Food Garden for nearly 40 years (established in 1983). The demonstration site grows 200–300 species and cultivars annually and features an outdoor teaching classroom and extensive interpretive signage. Sustainable urban agriculture is practiced (e.g., composting, crop rotation, green manures, pollinator plantings, no applications of synthetic pesticides or herbicides, etc.). From June to October, Botanical Garden staff, students and volunteers collaborate to grow, harvest, prepare and share the food. Annually, ~1000 lbs of fresh fruits and vegetables are distributed to local food security programs, enhancing the amount of nutrients these programs are able to provide.

Engagement programs work towards strengthening community capacity to produce and consume healthy food year-round (2.4).

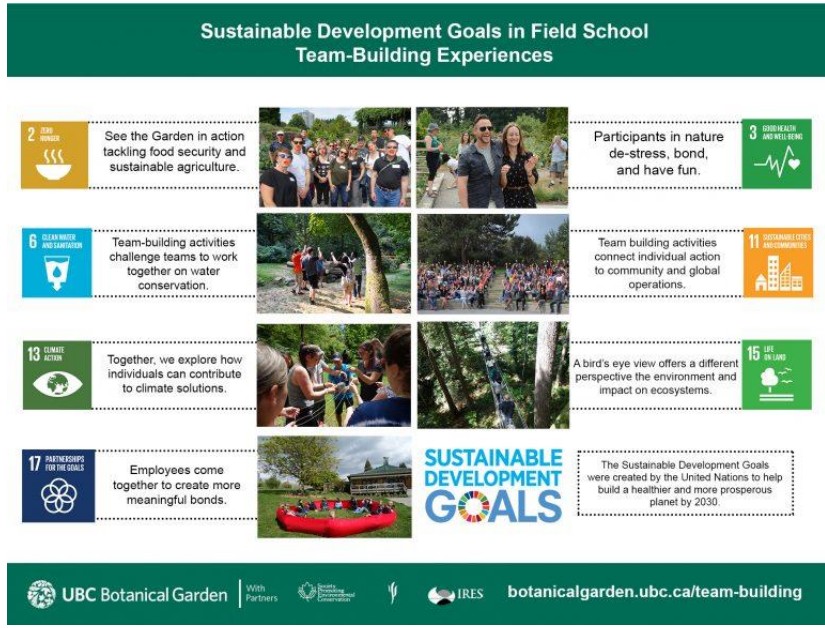

**Figure 2.** The Sustainable Communities Field School was established in 2015 with a focus on connecting businesses and teams with sustainability (UN-SDGs), climate action and pro-environmental behaviors. From 2015–2020, the program engaged ~90 teams and reached over 2500 participants.

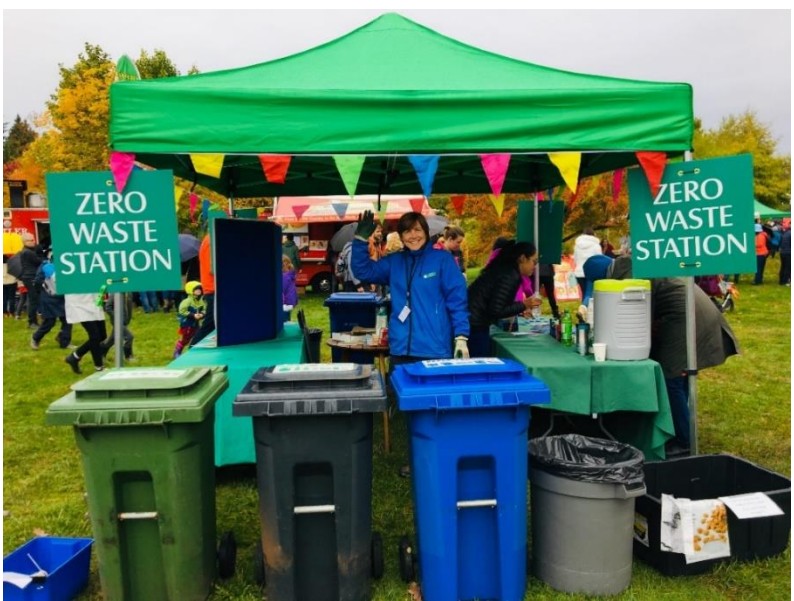

**Figure 3.** UBCBG's Zero Waste Station (Goal 12, Target 12.5) during our annual Apple Fest in 2019. As part of the Field School program, staff coordinate volunteer-run waste stations to reduce waste and promote zero waste education. Celebrating its 30th anniversary in 2021, Apple Fest showcases apple diversity through apple sales, tastings and entertainment.

As part of a collaborative network to advance North American Crop Wild Relative Conservation, UBCBG is working toward building capacity for conserving food plant diversity and crop wild relatives across North America (2.5) by developing networks and advancing strategic approaches for collaborative conservation [28]. Recent efforts to

inventory crop wild relatives across Canada using data from Botanical Gardens and other institutions provides a baseline of genetic resources available [29].

*3.5. Goal 4—Quality Education (Two Targets by UBCBG)*

Accessible learning is essential for plant conservation, and botanical gardens are important players in this area. For UBCBG, two targets for Goal 4 were identified (4.4 and 4.7). Towards Target 4.4, UBCBG's Horticulture Training Program equips students with technical skills for employment in horticulture, agriculture and landscaping. Over its 10-year history, the program has graduated close to 150 students (15 students per cohort).

Target 4.7 outlines the importance of knowledge and skills needed to promote sustainable development. This target is connected to UBCBG's diverse educational activities, such as garden experiences, interpretive signage and integration of Indigenous languages into staff training and garden educational experiences. The section of this target that refers to the promotion of a culture of peace and non-violence, global citizenship and appreciation of cultural diversity and of a culture's contribution to sustainable development, was connected to the work of Nitobe Memorial Garden. The Nitobe Memorial Garden is an important site for engaging visitors in learning about cultural diversity and Japanese–Canadian history. This garden showcases Japanese Garden design principles that espouse connections with the natural world.

> *"These are always in my mind when I take care of Nitobe Garden: Respect and follow the nature, care for the space (entire atmosphere, space between branches etc), care for not only plants, also water, light, air, sound, stones etc., and care for not only shapes, also its density".*
>
> *Ryo Sugiyama, Curator, Nitobe Memorial Garden, speaking to the uniqueness of Japanese horticultural practices.*

*3.6. Goal 6—Clean Water and Sanitation (Two Targets by UBCBG)*

For UBCBG, conserving water use has always been a significant challenge. The garden receives ~1200 mm of precipitation annually and has a climate moderated by its proximity to the Pacific Ocean. However, most of the precipitation falls between October and April, and the shallow, fast-draining soils are quickly depleted of moisture as temperatures warm in late spring. Irrigation comes with many challenges, including the use of appropriate technology and ongoing maintenance of old infrastructure.

In some areas of the garden, our focus is on nurturing sustainable landscapes, such as in the Garry Oak Meadow Garden, a representation of a locally threatened rain-shadow ecosystem where plants are adapted to open, sunny slopes and meadows. A high proportion of the plants are deep-rooted trees and shrubs, grasses and summer-dormant geophytes that need little or no irrigation once established. Garden staff pay particular attention to maintaining habitats and microhabitats to sustain biodiversity. Staff make use of shade and moisture-retaining coarse woody debris and organic mulches to reduce the need for irrigation. While there are no naturally occurring permanent waterbodies on campus, because of the nature of the soil, we make every effort to slow down and retain the water that passes through the Garden. Interpretive signage in this garden highlights the role drought plays in our local ecology and encourages visitors to adapt water-efficient plantings and practices. Educational resources provided through online tools (e.g., Grow Green Guide with Metro Vancouver) help local governments and communities to consider water efficient gardens (6.b).

Without irrigation, many plants in the collections would be irreparably damaged or lost entirely. Although there are sections of the garden where the strategy has been to plant drought-resistant and summer-drought-adapted species, many of the collections require considerable summer moisture, including our collection of rhododendrons, maples, ferns, and other exotic plants in the David C. Lam Asian Garden. Most are mountain-dwelling plants adapted to cool temperatures, humidity and a summer monsoon—quite the opposite of Vancouver's summer conditions. Beginning in 2017, with the support of the

Franklinia Foundation, UBCBG replaced and expanded its leaky, overly complicated, and outdated irrigation system in the Asian Garden with a state-of-the-art irrigation control system designed for more efficient water delivery (6.4). The new system provides access to real-time data for climate change research through a centrally controlled network which can monitor changes through a network of sensors.

*3.7. Goal 13—Climate Action (Two Targets by UBCBG)*

In 2021, the UBCBG launched a 5-year program to continue its climate education work as well as to advance climate adaptation planning. This aligns with Target 13.2, to integrate climate change measures into policies, strategies and planning. In alignment with Target 13.3 (engaging youth in climate action), UBCBG staff participated as facilitators in UNLEASH—a global social innovation lab committed to bringing youth together to create solutions for the UN-SDGs. In 2019, along with 100 youth and 20 facilitators, a UBCBG staff member and student spent six days at the Fairy Lake Botanic Garden (Shenzhen, China) working on Goal 13—Climate Action (Figure 4). In 2020 and 2021, UBCBG staff remotely facilitated two virtual UNLEASH hackathons (short-term events designed to solve problems) in India. These hackathons specifically engaged youth in thinking through solutions for the UN-SDGs within the context of a global pandemic.

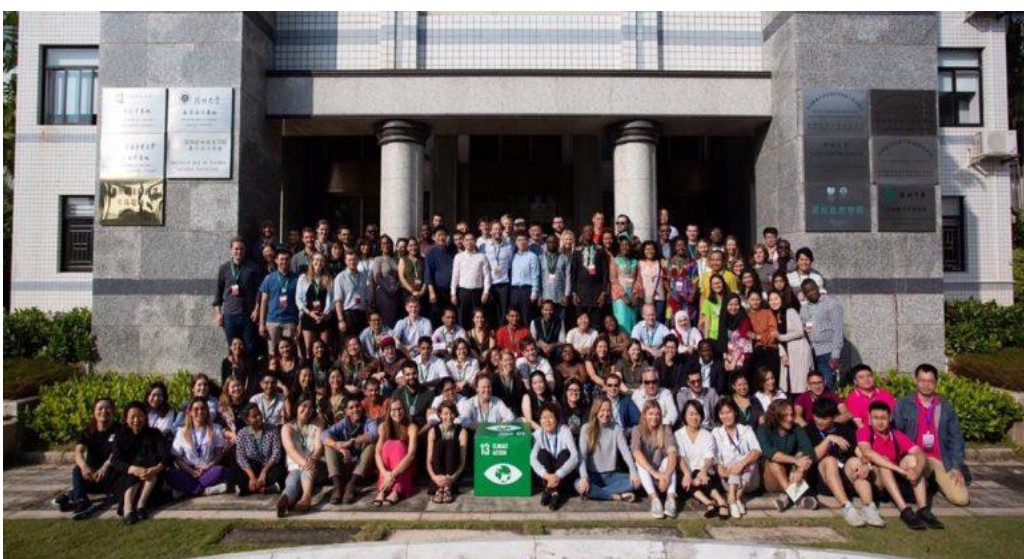

**Figure 4.** In November 2020, a UBCBG staff member and graduate student spent an intensive week at the Fairy Lake Botanic Garden in Shenzen China working with 100 youth to develop and design solutions for UN-SDG Goal 13—Climate Action. Following a five-step innovation process, youth are guided through a collaborative process to identify community issues and develop solutions that align to the UN-SDGs.

*3.8. Goal 17—Partnerships for the Goals (Two Targets by UBCBG)*

Botanical garden networks operate across local, regional, national and international boundaries and are well positioned to support partnerships for sustainable development goals (17.16). Given the magnitude and scale of action needed to achieve the UN-SDGs, partnerships across all scales are necessary. As a university botanic garden, there are direct connections to the university's sustainability initiatives, as well as expertise across diverse disciplines. UBCBG also partners with local and regional governments in the Metro Vancouver area in the promotion of sustainable landscapes, lawn alternatives, native plants, urban forestry and pollinator friendly practices (17.17).

### 3.9. Other UN-SDGs Advanced by UBCBG

UBCBG advanced single targets for several UN-SDGs. For instance, Goal 3—Good Health and Wellbeing (3.4) was linked to tours hosted in the garden, which promote mental health and well-being through facilitated activities. A recent partnership with medical groups across Canada on the Prescription for Nature (PaRx) enables patients to gain free access to UBCBG with a prescription from their doctor. Towards Goal 7—Affordable and Clean Energy, UBCBG's horticulture team is transitioning its operations towards more efficient energy usage through retrofitting and a transition to battery-electric machinery (7.3). For Goal 8—Decent Work and Economic Growth, UBCBGs contributions to hosting tourists onsite as well as collaborating with attractions across Vancouver are actions promoting sustainable tourism (8.9). The UBCBG Community Outreach Program provides free access and programming for low-income families, and access to UBCBG is free for all Indigenous Peoples (Goal 10—Reduced Inequalities (10.2)). The Garden contributes towards Goal 11—Sustainable Cities and Communities (11.3) by participating in UBC biodiversity advisory groups that promote plant and biodiversity conservation in urban design.

### 3.10. UN-SDGs Not Being Advanced by UBCBG

There were five UN-SDG goals for which no UBCBG contribution was identified. These goals included 1—No Poverty, 5—Gender Equality, 9—Industry, Innovation and Infrastructure, 14—Life below water and 16—Peace, Justice and Strong Institutions. These goals and targets highlight the gaps in UBCBG's sustainability efforts while presenting potential opportunities for UBCBG to consider in future programs and activities.

## 4. Discussion

This paper provided a valuable opportunity for the UBCBG team to collectively learn more about the UN-SDGs and to reflect on progress and gaps in sustainability.

Results of the assessment found that UBCBG is contributing to 12 of the 17 UN-SDGs. We listed 32 activities of the UBCBG that are advancing 24 of the 169 SDG targets. Goal 15—Life on Land and Goal 12—Responsible Consumption and Production are particularly relevant, with five and four targets respectively being addressed. Goals 2, 4, 6, 13 and 17 each had two targets being advanced. For Goals 3, 7, 8, 10 and 11, one target was advanced. The UN-SDGs not currently being addressed through UBCBG's work include Goals 1, 5, 9, 14 and 16.

Even though the literature on the role of botanic gardens in implementing SDGs is scant, this work shows that gardens can align their actions to the UN-SDGs since their work is closely tied to plant conservation, climate action, public education and sustainability. By reviewing the UBCBG alignment towards the UN-SDG, this paper contributes to both academic literature and practical information about the growing role of botanic gardens in sustainable community development.

Our results highlight a diversity of UBCBG activities that align to the UN-SDGs as well as highlight gaps in our work and priorities for future consideration. For UBCBG, this assessment provides an important snapshot of where we are in relation to the UN-SDGs. Additional work is needed to further prioritize actions and identify strategic directions. The UN-SDGs are just one reference point to guide plant conservation and human well-being. There are many other local, regional and international policy frameworks that botanical gardens might align to. As multifunctional spaces, it can be difficult for botanic gardens to determine which frameworks are most relevant. For example, plant-focused policies, such as the Global Strategy for Plant Conservation [30], but see [31], may have more practical metrics and indicators for ex situ conservation activities. However, delays in global policies (e.g., Global Post 2020 Framework) have slowed progress. In most cases, gardens need to prioritize actions to serve their communities (local, regional, national and global).

Tracking the progress of botanic gardens requires indicators that can help with assessing impact. While evaluating UBCBG's contributions to the UN-SDGS, we found it

difficult to evaluate UBCBG activities without appropriate indicators. This was a limitation to our study. For example, actions that may have a large impact and have been built on over several years, were similarly weighted as actions that may have a smaller impact, such as a single event or short-term project. By identifying potential metrics for UBCBG to explore (Table 1), we aim to contribute to future tracking and benchmarking efforts of botanic gardens. The Garden's endeavor to define indicators for botanical gardens can build off previous efforts of botanical gardens as well as learning from other sectors such as businesses and universities [32].

Goal 15 is considered to be the most relevant goal to the conservation work of botanic gardens [14]. UBCBG holds more than 5000 distinct taxa from 200 plant families from around the world and partners globally with other botanical gardens on plant conservation and plant collections management, strengthening conservation efforts. For example, as leaders of the Global Consortium for Acer, UBCBG supports strategies for global ex situ efforts. The Alpine Garden is an important site for engaging and educating the public about the fragile nature of alpine ecosystems, however increasing partnerships and funding could leverage the development of more interpretation signage, an alpine conservation curriculum and in situ conservation projects lead by garden staff [27]. In the food garden, UBCBG grows a diverse selection of food plants and promotes local food security while also promoting conservation of crop wild relatives and food plants important to Indigenous Peoples.

Across UBCBG operations and horticulture, several measures have been taken towards responsible consumption and production of energy, water, waste and resources. However, we lack an integrated plan of action, which means we are not able to measure, track and improve efficiencies across all the garden areas. An energy, water and resource audit for UBCBG would enable us to better understand how to minimize consumption. By planning sustainable landscapes, the Garden reduces water use and promotes sustainable biodiversity and ecosystem services. UBCBG staff practice innovative horticultural practices that integrate natural processes (e.g., leaving coarse woody debris on the ground and retaining standing dead timber ("snags")) and highlight to our community practices that support beauty and function. Future research and public engagement related to sustainable horticulture practices could support others in our community to adopt new techniques.

Sustainability education aligned to UN-SDGs offers a rich platform for public education. Botanic gardens play a key role in making people aware of the plant world and their importance for life on earth. Plant blindness hinders the implementation of SDGs [33,34] and botanical gardens with their living collections, expertise in horticulture, plant science and public education have a major role to play. Amprazis and Papadopoulou (2020) highlights that ignoring flora can skew systemic thinking and impede the success of several SDGs. As public spaces that attract people from diverse walks of life, botanic gardens can play an important role as hubs for place-based holistic learning that encompass the themes of the SDGs.

Several articles also point out the paradox of UN-SDGs: they focus on inclusive economic growth but ignore the realities and power structures that increase inequalities and pressure on natural resources, exacerbating biodiversity loss, climate change and resulting social tensions [35–37]. Koppnina suggests alternative education that emphasizes planetary ethic and degrowth as well as education on alternative paradigms such as Indigenous learning, circular economy, and liberation. Botanical gardens are often seen as leaders in environmental education and by incorporating alternative worldviews and systemic thinking into their educational programs, they may sow the seeds for these concepts to germinate in the broader community.

At UBCBG, education comes in many forms and reaches a diversity of audiences. The UBCBG Horticulture Training Program provides quality education and prepares students for careers in sustainable horticulture. Garden tours are a popular program at UBCBG. In 2019, UBCBG hosted ~250 groups (5000–6000 participants) on guided garden experiences that highlighted local biodiversity and sustainability themes. Long-established events like

the UBC Apple Festival promote food plant diversity and local farmers. Operating since 1991, Apple Fest has been organized by the volunteer group, Friends of the Garden (FOGs). The FOGs have also been crucial to maintaining other activities such as the Shop in the Garden, the Plant Centre and the UBCBG Hortline (a horticultural service that responds to public inquiries about plants and gardens). Sustainability education involves understanding human behaviour and learning. Practical resources, such as the *Behavior Change for Nature: A Behavioral Science Toolkit for Practitioners*, have been helpful while designing sustainability programming [38]. The UN-SDGs were first introduced to UBCBG through the Field School with educational outcomes aimed at prompting sustainable behaviours for food choices, water conservation, waste reduction and biodiversity stewardship. However, designing garden-based experiences that support and lead to sustainability action is complicated to monitor and predict.

Collaborations have been key to advancing sustainability at UBCBG. For example, through a partnership with Metro Vancouver (regional government), we co-host the Grow Green Guide. This online tool aims to support new gardeners in the region and features over 100 designs for urban spaces [17]. In 2020, the Grow Green team shifted from public outreach events (cancelled due to COVID-19) to hosting livestream webinars that reached a broad audience across the region (~130,000 reached over 2020–2021). As part of UBC, the Garden collaborates with researcher co-supervising undergraduate and graduate students on projects exploring topics such as behaviour change, citizen science, climate change and ex situ conservation. We also provide plant material for botanists and other researchers as well as supporting citizen science initiatives such as monthly bird counts on e-Bird. Our involvement with UBC Campus Biodiversity Initiative: Research and Demonstration (CBIRD) helps mainstream biodiversity conservation on campus through student-led research. The results point to important areas for potential growth.

Results of the UBCBG SDG assessment show that Goals 1, 5, 9, 14 and 16 are not currently being advanced; also, there are many other targets within other goals that UBCBG is not addressing. These goals and targets need to be considered, especially as UBCBG embarks on a multi-year initiative to advance climate adaptation planning and climate education. Not all UN-SDGs goals and targets will be relevant to UBCBG, but ensuring that our programs are inclusive and 'leave no one behind' will be key to serving our communities and making sure that the wellbeing benefits of being out in nature are accessible to all.

UBCBG's main activities are in horticulture, ex situ conservation and public education. This assessment of UBCBG contributions against the UN-SDGs is a key step towards understanding future direction. Additional work is needed to refine strategies to meet the urgency of the biodiversity crisis, climate emergency and social inequality. This assessment highlighted the need for a cohesive strategic plan for UBCBG. A collaborative plan can provide direction and help identify resources that will be necessary to implement the plan, proving the effectiveness of our activities. Specific opportunities for UBCBG's future action towards UN-SDGs were identified in this assessment process. While not a complete list, these potential actions can and should be considered as part of a larger strategy (Table 2).

**Table 2.** Gaps in programming and activities, and potential future actions for UBCBG identified in our survey to staff across departments.

| UN-SDG | Future Actions Identified in our Survey |
| --- | --- |
| 2. Zero Hunger | Further incorporate Indigenous foods, human health and nutritional information into the planning of plantings of the food garden and education programming. |
| 3. Good Health and Well Being | Increase access to the Garden for all members of the community. Programming outside the Garden to connect people to nature. |
| 4. Quality Education | Increase the reach of our educational programs to serve low-income communities and students. Extend garden programs outside the garden and within communities. Design educational programs that help the public develop an in-depth understanding of ecology and threats to our ecosystems. |
| 5. Gender Equality | Increase diversity within Garden staff and leadership positions. |
| 6. Clean Water and Sanitation | Increase water conservation infrastructure and education across the garden. |
| 7. Affordable and Clean Energy | Collaborate with UBC to move operations towards carbon neutrality to meet the 2030 Climate action of UBC, which aims at meeting a 85% reduction targets. |
| 8. Decent Work and Economic Growth | Promote youth employment and training across UBCBG's departments. |
| 9. Industry, Innovation and Infrastructure | Further advance research and innovation through citizen science, garden-based research and collaboration with UBC faculty. |
| 10. Reduced Inequality | Work in line with the UBC Indigenous Strategic plan to increase action for the advancement of Indigenous Peoples' human rights. Align UBCBG efforts to the UBC Inclusion Action Plan and the Wellbeing Strategic Framework. Increase participation from leadership positions at the Garden in the IDEA group. |
| 11. Sustainable Cities and Communities | Raise awareness of the importance and value of biodiversity in urban design, planning and development. Disseminate garden horticultural expertise to local and regional government and businesses. Increate outreach efforts to promote sustainable living. |
| 12. Responsible Consumption and Production | Advocate and demonstrate responsible consumption and production by reducing consumption of goods and identifying opportunities to promote sustainable consumption. Internal UBCBG energy and resource audit and reduction plan. |
| 13. Climate Action | Identify local climate actions and climate justice needs, and work with our networks to build capacity of communities to respond to climate change and biodiversity loss. |
| 14. Life Below Water | Advocate for conservation of land-water ecosystems to better protect local waterways. Emphasis on terrestrial and ocean ecosystem interactions in educational programming since the Garden is situated so close to the ocean. |
| 15. Life on Land | Define and expand conservation efforts for UBCBG to determine opportunities and capacity to advance both ex situ and in situ conservation. Mobilize financial support for conservation, sustainability and infrastructure needs. |
| 17. Partnerships to achieve the Goal | Leverage collaborations to initiate on the ground action towards conservation. Work with government and other partners to mainstream biodiversity conservation and climate action in decision-making and policy. |

## 5. Conclusions

The UN-SDGs provided us with a framework to understand and evaluate our contribution to sustainability. We are re-imagining the role of botanical gardens in an age of equity, decolonization, the biodiversity crisis and the climate emergency. This process of evaluation highlighted three main priorities to improve our effectiveness in meeting sustainability goals and addressing global environmental threats in an intentional and meaningful way.

The first priority is developing a collaborative (involving garden staff and partners) strategic plan. This process would include evaluating our effectiveness, identifying ways to increase impact to further our mandate and raise funds and resources needed to meet goals identified in the strategic plan. Inclusion, diversity and equity are essential in strategic planning to ensure that no-one is left behind.

Within the garden, we need to demonstrate and inspire sustainability. An energy, water and resource audit and plan could help to reduce consumption and make operations more sustainable and further demonstrate our commitment to sustainability.

Finally, we need to go back to our roots and be inspired by the Garden's history. The founder John Davidson was deeply involved in the community on the ground. He set up several school gardens, studied native plants and advocated for the preservation of local forests and watersheds. Our involvement in addressing the biodiversity crisis outside the garden needs to be scaled-up by all garden departments–conservation, horticulture, research and education. This would mean on-the-ground action for conservation and education, as well as mainstreaming climate and biodiversity in policy and decision-making.

Since the UN-SDGs address both nature and people, they are an appropriate framework to guide our work. With only eight years left to 2030, the UN-SDGs need our urgent attention now.

**Supplementary Materials:** The following supporting information can be downloaded at: https://www.mdpi.com/article/10.3390/su14106275/s1, Supplementary File S1: Template for mapping UN-SDGs to Garden's programs and activities.

**Author Contributions:** Conceptualization, T.M., D.B. and A.L.-V.; methodology, T.M., D.B. and A.L.-V.; writing—original draft preparation, T.M., D.B. and A.L.-V.; writing—review and editing, T.M., D.B., A.L.-V., P.L., D.A., J.D., L.C., A.H., B.S., K.K. and R.S.; visualization, A.L.-V. All authors have read and agreed to the published version of the manuscript.

**Funding:** UBCBG acknowledges the generous financial support from donors for the Sustainable Communities Field School and from the Franklinia Foundation for the Preservation and Expansion of the David C. Lam Asian Garden.

**Institutional Review Board Statement:** Not applicable.

**Informed Consent Statement:** Informed consent was obtained from all subjects involved in the study.

**Data Availability Statement:** Not applicable.

**Acknowledgments:** The authors acknowledge the ancestral and unceded territory of the Musqueam People where UBC Campus and the UBCBG are located. The authors would like to thank Daniel Mosquin and Douglas Justice whose comments improved the manuscript. We also acknowledge support from the Faculty of Science and volunteer contributions from the Friends of the Garden. Our gratitude to the staff, students and horticulturalists who steward the Garden.

**Conflicts of Interest:** The authors declare no conflict of interest. The funders had no role in the design of the study; in the collection, analyses, or interpretation of data; in the writing of the manuscript, or in the decision to publish the results. All the authors of this manuscript were affiliated to UBCBG at the time of the development, preparation, and publication of this study. Authors made all possible efforts to be objective and impartial when evaluating UBCBG's programs and activities.

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
