# Peer review of "Aligning to the UN Sustainable Development Goals: Assessing Contributions of UBC Botanical Garden"

_sustainability, doi:10.3390/su14106275_

Round 1
Reviewer 1 Report
The revised paper is an insightful and needed study on reshaping the role of botanical gardens in an age of climate emergency for both nature and people. Perhaps this changing role highlighted by the authors at the end of the article should be more emphasized in the introduction and abstract. There are no conclusions in the paper's structure. In practice, however, they are included in the section named by the authors as a discussion. At the same time, the missing part of the article is precisely the discussion. The presented descriptive content could be supplemented with quotes from the interviews in the results section. At the bottom of page number five (12.2 Potential UBCBG Metrics), instead, "Units of plastic and peat used", it should probably be "pots made of more robust material that can be reused and ultimately recycled".

Author Response
We respond to specific comments, point-by-point below.
Reviewer #1
- The revised paper is an insightful and needed study on reshaping the role of botanical gardens in an age of climate emergency for both nature and people. Perhaps this changing role highlighted by the authors at the end of the article should be more emphasized in the introduction and abstract.
Response: We thank the reviewer for the insightful comments. We have now emphasized more the reshaping role of botanical gardens in an age of climate emergency for both nature and people in our abstract (lines 19-22) and introduction (lines 33-42).
- There are no conclusions in the paper's structure. In practice, however, they are included in the section named by the authors as a discussion. At the same time, the missing part of the article is precisely the discussion. The presented descriptive content could be supplemented with quotes from the interviews in the results section.
Response: We thank the reviewer for pointing this out. We have now added a “Conclusions” section in line 501 and have improved our Discussion based on the reviewer’s suggestion. Given the lack of peer-reviewed literature on the contribution of Botanical Gardens to sustainability, and in particular to the UN-SDGs, we have focused our attention into putting our results in context with three main areas: the conservation of biological diversity, UN-SDGs in education and businesses (e.g. lines: 395-397, 403-406, 422-429, 431-439, 450-452). In addition, we have incorporated some relevant answers from our interviews with staff on Table 2 (line 493). These answers highlight the staff’s perception from each UBCBG department, gaps in the contribution to each UN-SDG goal (we currently contribute to), the identification of future action and the limitations to achieve them.
- At the bottom of page number five (12.2 Potential UBCBG Metrics), instead, "Units of plastic and peat used", it should probably be "pots made of more robust material that can be reused and ultimately recycled".
Response: Thank you. We have now changed the wording to read “pots made of more robust material that can be reused and recycled".
Reviewer 2 Report
A bit more explanation is needed for the section describing Potential UBCBG Metrics (table 1). Most of them are absolute and only few (7.3, 15.1) are relative measures. In my opinion, it does not make sense to use absolute values such as number of YT video views or number of activities. Especially, that a number of activities or programs does not necessarily mean they're effective, relevant or sustainable.
To the section describing the methods, in my view, it is necessary to add a disclaimer, presenting what is the impact of the authors' place of work on the choice of the object of research and indicators (I do not want to imply results). The reader may have the impression that this type of self-evaluation conducted will serve to justify the mission and positive functions of the institution employing the authors. There is nothing inappropriate about doing research on one's own academic workplace, but it is important to be aware of this and the possible bias.
In this sense, it would also be appropriate to expand the description of the "conflicts of interest" section, by adding that the researchers are studying UBCBG, which is related to their academic affiliation, but that they have made efforts related to objectivity
Author Response
Reviewer #2
We respond to specific comments, point-by-point below.
- A bit more explanation is needed for the section describing Potential UBCBG Metrics (table 1). Most of them are absolute and only few (7.3, 15.1) are relative measures. In my opinion, it does not make sense to use absolute values such as number of YT video views or number of activities. Especially, that a number of activities or programs does not necessarily mean they're effective, relevant or sustainable.
Response: We thank the reviewer for pointing this out. We have now revised the proposed metrics and have changed absolute values for percentages or proportions whenever possible. We have also added a sentence in the caption of Table 1 that reads as follows: “The proposed metrics are expressed in relative values as these are more effective and relevant for internal evaluation purposes, however in some cases absolute values are included.”
- To the section describing the methods, in my view, it is necessary to add a disclaimer, presenting what is the impact of the authors' place of work on the choice of the object of research and indicators (I do not want to imply results). The reader may have the impression that this type of self-evaluation conducted will serve to justify the mission and positive functions of the institution employing the authors. There is nothing inappropriate about doing research on one's own academic workplace, but it is important to be aware of this and the possible bias.
Response: We agree with the reviewer comment, and we have now added the following sentences to lines 138-141 of the Methods section to address the issue of potential bias on the impact of the authors’ place of work and the object of study. “This study was a self-evaluation of the garden against the SDGs conducted by staff at the UBCBG and may inherently contain some bias. However, we tried to be as impartial and objective as possible and aimed to follow a SWOT assessment of the Strengths, Weaknesses, Opportunities and Threats to provide as holistic a review as possible.”
- In this sense, it would also be appropriate to expand the description of the "conflicts of interest" section, by adding that the researchers are studying UBCBG, which is related to their academic affiliation, but that they have made efforts related to objectivity
Response: The section of “Conflict of Interest” now mentions the following statement: “All the authors of this manuscript were affiliated to UBCG at the time of the development, preparation, and publication of this study. All authors made all possible efforts to be objective and im-partial when evaluating UBCBG’s programs and activities.”
Reviewer 3 Report
This manuscript is very interesting for people interested to learn about UN Sustainable Development Goals and the contribution of botanical gardens. However, my comment is that the manuscript only shows the positive and stronger sides of the University of British Columbia Botanical Garden (UBCBG) and does not describe much about future actions. For a balanced and critical analysis, it might be a good idea to add a paragraph to address: Which are the key Goals, Targets, or Indicators in which the UBCBG requires further attention and action?
Author Response
We respond to specific comments, point-by-point below.
Reviewer #3
- This manuscript is very interesting for people interested to learn about UN Sustainable Development Goals and the contribution of botanical gardens. However, my comment is that the manuscript only shows the positive and stronger sides of the University of British Columbia Botanical Garden (UBCBG) and does not describe much about future actions. For a balanced and critical analysis, it might be a good idea to add a paragraph to address: Which are the key Goals, Targets, or Indicators in which the UBCBG requires further attention and action?
Response: We agree with the reviewer’s and have now taken two approaches to address this point. First, we have now improved our Discussion to put our results in context with research on UN-SDGs in the areas of conservation of biological diversity, education and business, highlighting aspects that could be further address to advance UN-SDGS (e.g. lines: 395-397, 403-406, 422-429, 431-439, 450-452). Secondly, we have incorporated the answers from our interviews with staff to Table 2 (line 493) addressing the three following questions aiming at identifying the staff’s perception from each UBCBG department, gaps in the contribution to each UN-SDG goal (we currently contribute to), the identification of future action and the limitations to achieve them. The questions we asked were:
Q1. Which are the key Goals, Targets, or Indicators in which the UBCBG requires further attention and action?
Q2. What aren’t we doing that we could be doing more (focus on each individual staff areas and SDG previously identified)?
Q3. What are the limitations to achieve that?